# Response of Winter Wheat to Delayed Sowing and Varied Nitrogen Fertilization

Wacław Jarecki

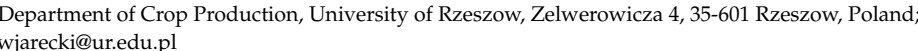

Department of Crop Production, University of Rzeszow, Zelwerowicza 4, 35-601 Rzeszow, Poland; wjarecki@ur.edu.pl

**Abstract:** Common wheat is one of the most important cereal crops in the world. In cultivation, winter, spring, and facultative varieties of this species are known. In wheat agronomy, timely sowing and optimal nitrogen fertilization are particularly crucial practices, as both significantly impact yield and grain quality. In a three-year field experiment, the response of the winter wheat variety RGT Kilimanjaro to two sowing dates (recommended and delayed by 30 days) and varied nitrogen fertilization levels (100 kg ha$^{-1}$, 150 kg ha$^{-1}$, and 200 kg ha$^{-1}$) was investigated. It was shown that the difference in grain yield between 2021 and 2023 amounted to 0.74 kg ha$^{-1}$. The application of 200 N kg ha$^{-1}$ significantly increased the Soil Plant Analysis Development (SPAD) index and Leaf Area Index (LAI) compared to the variant with a delayed sowing date and a nitrogen dose of 100 kg ha$^{-1}$. Yield components (number of spikes per square meter, thousand grain weight) and grain yield were highest when wheat was sown at the recommended date and with the application of either 150 or 200 N kg ha$^{-1}$. The number of grains per spike significantly varied between the variant with the recommended sowing date and a dose of 200 N kg ha$^{-1}$ and the variant with a delayed sowing date and a dose of 100 N kg ha$^{-1}$. The lowest grain yield was obtained at a 30-day late wheat sowing date when applying 100 N kg ha$^{-1}$. The protein content in the grain was primarily influenced by nitrogen fertilization. Therefore, it can be concluded that delaying the sowing date of winter wheat by 30 days results in a decrease in grain yield but can be compensated by increased nitrogen fertilization. The most favorable economic effects were achieved with the application of 150 N kg ha$^{-1}$ at the recommended sowing date. Considering that high doses of nitrogen fertilization can have adverse effects on the natural environment, research in this area should be continued.

**Keywords:** *Triticum aestivum* L.; sowing date; fertilization; nitrogen; yield components; yield; protein

## 1. Introduction

Common wheat (*Triticum aestivum* L.) is one of the most important crops in the world, as it ensures food security for humans and serves as fodder for livestock [1]. Therefore, agricultural research continually improves varieties [2,3] and cultivation technology [4] of this species to achieve stable, high-quality grain yields. A study by Neupane et al. [5] has indicated that as a result of climate change, many crops may show reduced yields, especially in regions with insufficient precipitation. Dueri et al. [6] have confirmed that climate changes can reduce cereal yields, requiring adjustments to agronomic recommendations (e.g., sowing date, sowing density, etc.), such as those for winter wheat. These authors highlighted the optimal sowing date as a viable method for adapting this species to climate change and mitigating its effects. Oleksiak [7] has reported that despite the awareness of the significant importance of timely sowing of winter wheat, a large portion of plantations is sown at a delayed schedule in certain regions. In Poland, this is often attributed to the late harvest of preceding crops, such as grain maize or sugar beets. Qiao et al. [8] argued that the sowing date primarily determines the plants' response to photoperiodism and vernalization, which in turn affects the development of generative organs. When the sowing date is delayed, it results in a reduction in the growth and development of

plants. Research by Chu et al. [9] indicates that delaying the sowing of winter wheat has a significant impact on yield, its components, and grain protein content. A two-week delay in sowing in the latter study resulted in an increase in the number of grains per spike and overall yield. However, the protein content in the grain decreased due to the reduced nitrogen uptake by the plants. Ferrise et al. [10], on the other hand, showed that the protein content of wheat grain was significantly higher when sown at a delayed date compared to the recommended date. Research by Liu et al. [11] demonstrated that the yield of winter wheat decreased by approximately one percent with each day of delayed sowing compared to the optimal date in the study region. The decline in yields primarily resulted from slower plant growth, reduced yield components, and decreased nitrogen utilization from fertilizers. Additionally, the consequence of delayed sowing was the exposure of plants to low temperatures during vegetative growth, a shortened duration of individual developmental phases, and elevated temperatures during grain filling. Moghaddam et al. [12] concluded that recommending an earlier sowing date combined with optimal fertilization may be necessary to adapt wheat cultivation to climate change in dry areas. In the context discussed, Thorup-Kristensen et al. [13] stated that winter wheat is more efficient in nitrogen uptake from the soil than spring wheat, which has important economic and environmental implications. Bulut et al. [14] demonstrated that the efficiency of nitrogen and water utilization decreased with the delay of wheat sowing dates, but the increased grain sowing compensated for these losses. Dagash et al. [15] have demonstrated that the sowing date and nitrogen fertilization significantly influence wheat grain yield. However, the results of these authors, and in particular nitrogen fertilization, were dependent on weather conditions in individual seasons. Fu et al. [16] reported that delaying the sowing of winter wheat could reduce yields by up to 13.7%, while increased nitrogen fertilization mitigated yield losses. Dar et al. [17] confirmed that delaying the sowing of wheat resulted in a decrease in grain yield. However, the application of nitrogen at a rate of 100 kg ha$^{-1}$ exerted a favorable impact on plant growth, yield components, and overall yield. Further increases in nitrogen fertilization above 100 kg ha$^{-1}$ were unjustified, regardless of the sowing date. Brzozowska and Brzozowski [18] demonstrated that higher NPK fertilization had a significant impact only on the nitrogen content in the grain and did not compensate for the delay in wheat sowing. Mohamed et al. [19] proved that delaying wheat sowing significantly reduced the evaluated plant parameters, except for the thousand grain weight; however, all traits improved when nitrogen fertilization was increased from 75 to 125 kg ha$^{-1}$. Abedi et al. [20] showed that achieving maximum grain and protein yield in wheat required the application of 240 kg ha$^{-1}$ of nitrogen. However, they believed that excessive nitrogen application was economically unjustified and detrimental to the environment. Zheng et al. [21] shared a similar perspective, considering that an excessively delayed sowing date combined with high nitrogen doses was unfounded, including economic aspects.

From previous studies [22,23], it is known that nitrogen fertilization is one of the agronomic practices that has a significant impact on the quantity and quality of winter wheat yield. Tabak et al. [24] demonstrated that the optimal nitrogen dosage for winter wheat was 217 kg N ha$^{-1}$, resulting in a maximum grain yield of 8.25 t ha$^{-1}$. Khan et al. [25] argued that a sufficient nitrogen dosage for wheat was 140 kg N ha$^{-1}$. However, nitrogen should be divided into three doses at a ratio of 1:2:1 and applied on the following dates: before sowing and at the tillering and shooting stages. On the other hand, Litke et al. [26] demonstrated a significant increase in wheat yields with an increasing nitrogen dosage of up to 180 kg ha$^{-1}$. At the same time, an increase in the nitrogen application rate to 210 kg ha$^{-1}$ had a positive effect on grain quality indicators, with the exception of starch content. Ducsay and Ložek [27] proved that fertilizing with nitrogen to the level of 140 kg N ha$^{-1}$ positively influenced wet gluten and crude protein formation with the highest increment in the variant with nitrogen applied in the form of urea solution.

Klepeckas et al. [28] and Xu et al. [29] suggested that when conducting agricultural experiments, it is worthwhile to apply modern measurement techniques that allow for

a precise assessment of plant conditions and anticipated yields of species such as wheat. Yin et al. [30,31] highlighted the SPAD measurement as particularly useful, as it reliably assesses the nutritional status of plants during the growing season. Costa et al. [32] and Liu et al. [33] have concluded that the timing of agronomic practices, especially sowing dates, is dependent on the research region and experimental location. Therefore, they considered it necessary to conduct field experiments in various habitats to obtain reliable results and appropriate recommendations.

The aim of the study was to evaluate the response of winter wheat to two sowing dates (recommended and delayed by 30 days) and three nitrogen doses (100 N kg ha$^{-1}$, 150 N kg ha$^{-1}$, 200 N kg ha$^{-1}$). The research hypothesis assumed that a delayed sowing date would require an increased nitrogen dose to reduce yield losses.

## 2. Materials and Methods

The field experiment was set up at the Experimental Station of the University of Rzeszów in Krasne (50°03′ N 22°05′ E), near Rzeszów, Poland. The experiment was conducted in three growing seasons: 2020/2021, 2021/2022, and 2022/2023. The studied factors included the sowing date (recommended and delayed by 30 days) and various nitrogen fertilization doses (100 N kg ha$^{-1}$, 150 N kg ha$^{-1}$, and 200 N kg ha$^{-1}$). The selected variety for the trial was RGT Kilimanjaro (RAGT Semences Polska Sp. z o.o., Toruń, Poland), recommended for cultivation in the study area. The experiment was conducted in four replicates in a split-plot design. Weather conditions were compiled using data from the Meteorological Station of the University of Rzeszów, located approx 12 km from the experimental field. The experiment was established on medium soil, characterized by a slightly acidic pH (5.8–6.3 pH in KCl) and moderate humus content (1.1–1.3%). The content of available phosphorus (174–193 mg kg$^{-1}$ soil) and potassium (223–242 mg kg$^{-1}$ soil) was high, while magnesium (55–62 mg kg$^{-1}$ soil) was at a moderate level (Table 1). Soil samples were analyzed at the Regional Chemical and Agricultural Station in Rzeszów, according to Polish standards.

**Table 1.** Soil chemical analysis.

| Parameter | Unit | 2020 | 2021 | 2022 |
|---|---|---|---|---|
| pH in KCl | - | 6.3 | 6.2 | 5.8 |
| Humus | % | 1.3 | 1.1 | 1.2 |
| Phosphorus (P$_2$O$_5$) | | 193 | 184 | 174 |
| Potassium (K$_2$O) | mg·kg$^{-1}$ soil | 242 | 238 | 223 |
| Magnesium (Mg) | | 62 | 58 | 55 |

The forecrop was winter oilseed rape, and the field was disc-harrowed after its harvest. The recommended sowing density for the variety RGT Kilimanjaro is 350 seeds/m$^2$. Wheat grain was sown at a depth of 3 cm, with a row spacing of 12.5 cm. The surface area of a single plot was 16.0 m$^2$. Before sowing, a combined cultivator and NPK mineral fertilization were applied. The nitrogen (ammonium nitrate 34% N), phosphorus (superphosphate 19% P$_2$O$_5$), and potassium (potassium salt 60% K$_2$O) doses were 30, 60, and 90 kg ha$^{-1}$, respectively. Optimal (recommended) sowings were carried out on 28 September 2020, 26 September 2021, and 29 September 2022. Delayed sowings were conducted on 28 October 2020, 26 October 2021, and 24 October 2022. The following preparations were used for the chemical protection of plants: Expert Met 56 WG, Huzar Active Plus, Antywylegacz 725 SL + Moddus 250 EC, Boogie Xpro 400 EC, Karate Zeon 050 CS, and Fandango 200 EC. Plant protection products were used in accordance with the manufacturer's recommendations following prior monitoring of the plantation. Chemical sprayings were applied using a tractor-mounted sprayer. Plant development stages were determined according to the BBCH scale (Bundesanstalt, Bundessortenamt und Chemische Industrie) used in the EU [34]. In the spring, nitrogen fertilization (ammonium nitrate) was applied at two

dates: at the beginning of vegetation at a dose of 60, 80, or 110 N kg ha$^{-1}$, and at the stem shooting stage (BBCH 21) at a dose of 40, 70 or 90 N kg·ha$^{-1}$.

Leaf Area Index (LAI) was measured using an AccuPAR LP-80 apparatus (Meter Group, Inc., Pullman, WA, USA). Soil Plant Analysis Development (SPAD) was measured using a SPAD 502P chlorophyll meter (Konica Minolta, Chiyoda, Japan). The measurements of the LAI (m$^2$/m$^2$) and SPAD indices were carried out at the milk stage (BBCH 75) on 15 flag leaves in the morning.

The spike density, number of grains per spike, and TGW were counted for plants harvested from 1 m$^2$. The harvested yield (BBCH 89) was calculated per hectare with a grain moisture content of 14%.

The chemical composition of the grain (total protein) was determined using the near-infrared method and an MPA FT-LSD Spectrometer (Bruker, Mannheim, Germany) in the laboratory of the Department of Plant Production at the University of Rzeszów.

In the final stage, economic calculations were performed for the applied nitrogen fertilization. The calculations were based on market data from 2023.

The results were statistically analyzed using analysis of variance (two-way ANOVA) and Tukey's half-confidence intervals to determine the significance of differences between trait values. Statistical analysis was performed using TIBCO Statistica 13.3.0 (TIBCO Software Inc., Palo Alto, CA, USA).

## 3. Results and Discussion

Weather conditions varied across different seasons (Figure 1). In September 2020, rainfall was below the long-term average. On the other hand, in 2021, very low rainfall occurred in October. Significant variations in precipitation were observed in January compared to the long-term average, especially in 2023. From January to August 2023, precipitation levels were higher than the long-term average. On the other hand, in 2022, precipitation from May to August was below the long-term average. In the autumn months of each year, air temperatures were high or close to the long-term averages. Similarly, in the winter months, there were no temperature drops below the long-term average. In the spring and summer months, temperatures fluctuated both above and below the long-term average.

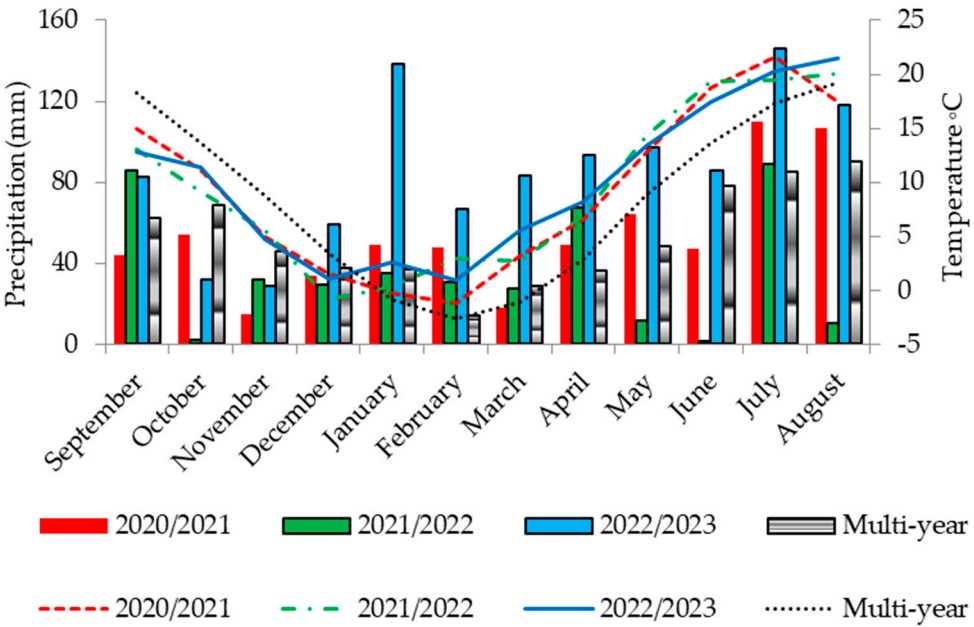

**Figure 1.** Weather conditions. The columns show the total precipitation, and the line shows the mean temperature.

Weather conditions often play a crucial role in affecting the studied parameters or traits of crops. The variations observed in these conditions across different study years can significantly influence the outcomes of long-term field trials. Additionally, the interaction between the tested factors and specific experimental years can further contribute to the variability of results in subsequent seasons [35,36]. Skowera et al. [37] have shown that in Poland, the highest risk of rainfall shortage occurs in June, a critical period as winter wheat enters the grain-filling stage from mid-June. Xu et al. [29] have reported that air temperatures play an important role in the proper growth and development of field crops, and their unfavorable patterns can be just as detrimental as a lack of precipitation. Yang et al. [2] have stressed that global warming and drought pose a serious threat to wheat production in many parts of the world, necessitating the breeding of new varieties and the improvement in agricultural technology. Khanna-Chopra et al. [38] have confirmed that wheat production is limited by heat stress. However, hexaploid species show good adaptation to such conditions, as evidenced by higher yields achieved compared to tetraploid or diploid species.

The Leaf Area Index (LAI) one side green area per 1 square meter of soil had the lowest value in the variant with delayed sowing and fertilization with a nitrogen dose of 100 kg ha$^{-1}$. Significantly higher LAI measurements were observed after the application of a nitrogen dose of 200 kg ha$^{-1}$ (Figure 2). The effect of variable nitrogen fertilization was modified by the sowing date, which was demonstrated by a significant interaction between the factors studied.

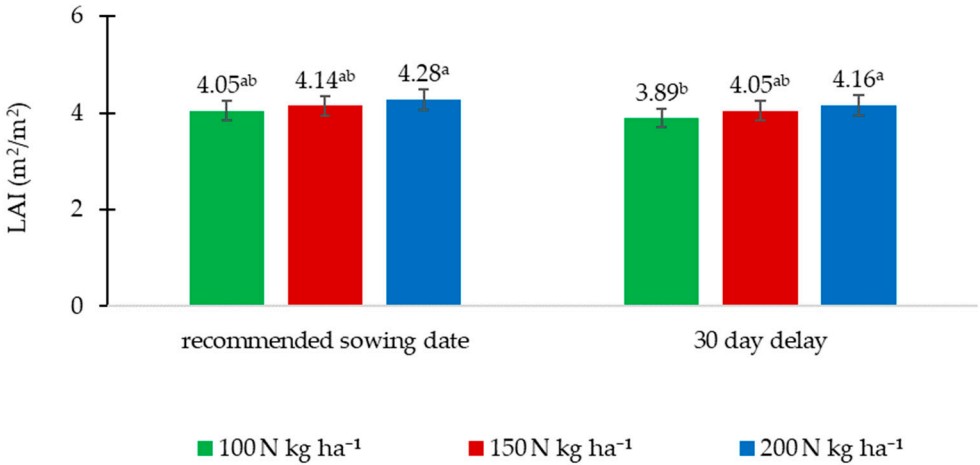

**Figure 2.** Measurement of the LAI index—interaction between the studied factors (mean over the years). Mean values with different letters in bars are statistically different ($p < 0.05$). The standard error is marked on the bars.

As expected, the SPAD index increased with the rise in nitrogen dose; however, significant differences were demonstrated only between the variants with a nitrogen dose of 200 kg ha$^{-1}$ and the variant with delayed sowing and a nitrogen dose of 100 kg ha$^{-1}$ (Figure 3).

Yin et al. [31] have shown that taking SPAD measurements on plants during the growing season allows for an increase in the range of results and evaluation of the effectiveness of the factors tested in experiments. This was also confirmed by my own results. Zhu et al. [39] proved that with delayed sowing of winter wheat, the LAI index and plant biomass was reduced. In my own research, this was generally just a trend.

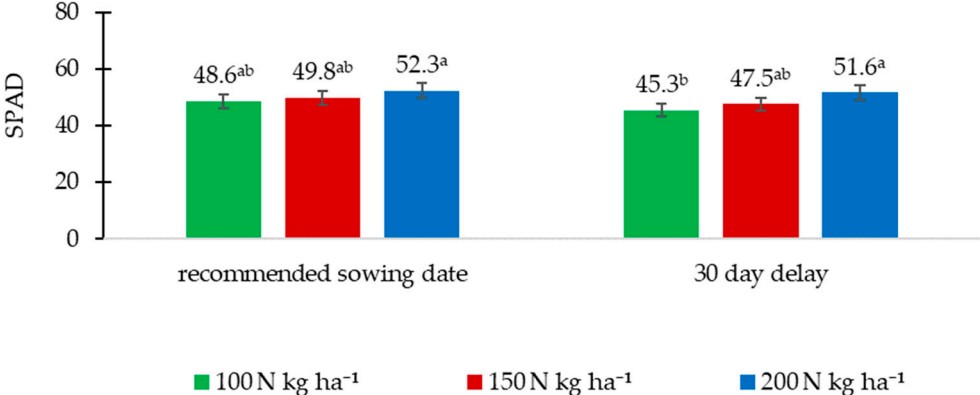

**Figure 3.** Measurement of the SPAD index—interaction between the studied factors (mean over the years). Mean values with different letters in bars are statistically different (*p* < 0.05). The standard error is marked on the bars.

Jarecki and Czernicka [40] confirmed the usefulness of field measurements of plant parameters, such as SPAD and LAI, in assessing the effectiveness of fertilization of winter wheat and the response of plants to the applied fertilizers. The aforementioned authors observed that SPAD and LAI indices in winter wheat could serve as predictors for plant growth and grain yield. In a study by Altaf et al. [41], it was reported that nitrogen fertilization, when combined with sulfur, had a positive impact on nitrogen accumulation, SPAD index, and plant physiology across diverse abiotic stress conditions. Nonetheless, the authors pointed out that plants exhibited increased sensitivity to high temperatures, particularly when concurrent environmental stresses were present.

On average, in the present experiment, the most favorable effect on the number of spikes per square meter and thousand grain weight (TGW) was achieved by sowing in the recommended period and applying nitrogen at a dose of 150 or 200 N kg ha$^{-1}$. The number of grains per spike showed significant variation only between the variant with recommended sowing date and a dose of 200 N kg ha$^{-1}$, compared to the variant with delayed sowing and a dose of 100 N kg ha$^{-1}$ (Table 2).

**Table 2.** The influence of the interaction of sowing date and dose of N on yield components (mean over the years).

| Sowing Date | Dose of Nitrogen (kg ha$^{-1}$) | Number of Ears (pcs/m$^2$) | Number of Grains per Ear | Thousand Grain Weight (g) |
|---|---|---|---|---|
| Recommended | 100 | 558.2 ± 5.36 ab | 29.7 ± 4.36 ab | 40.7 ± 3.47 ab |
| | 150 | 566.7 ± 6.25 a | 32.4 ± 5.62 ab | 41.9 ± 3.56 a |
| | 200 | 570.1 ± 8.36 a | 33.2 ± 5.83 a | 42.2 ± 4.41 a |
| 30-day delay | 100 | 545.4 ± 5.14 b | 27.3 ± 5.02 b | 38.4 ± 3.52 b |
| | 150 | 554.2 ± 5.69 ab | 31.2 ± 5.73 ab | 39.8 ± 3.84 ab |
| | 200 | 560.0 ± 7.37 ab | 31.8 ± 5.91 ab | 40.3 ± 4.62 ab |
| Mean | | 559.10 | 30.93 | 40.55 |

Results are expressed as mean value ± standard deviations. Mean values with different letters in columns are statistically different (*p* < 0.05).

Podolska and Wyzińska [42] showed that delayed sowing of winter wheat resulted in a decrease in the number of spikes and grains per square meter, but it did not affect the TGW index. Szumiło and Rachoń [43] reported that delaying the sowing date primarily reduced the number of spikes per square meter. Sobko et al. [44] demonstrated in their study that different sowing dates did not affect the number of grains per spike but significantly modified TGW. The latter authors obtained the lowest TGW after sowing wheat on October

10 or 20. It should, therefore, be stated that our own results and those of other authors are often ambiguous or even contradictory.

Donaldson et al. [45] demonstrated that the wheat yield was most influenced by the number of spikes per square meter and the number of grains per spike. However, with delayed sowing, they primarily observed a reduction in the number of spikes per square meter. Abdel Nour et al. [46] provided that the increase in nitrogen fertilizer dose positively correlated with wheat parameters such as the growing period, the number of spikes per square meter, thousand grain weight, and grain yield. Ducsay and Ložek [27], on the other hand, reported that increased nitrogen fertilization had a beneficial effect on wheat yield components, except for thousand grain weight. This was only partially confirmed in my study.

The grain yield of winter wheat was significantly different as a result of the interaction of the factors studied. On average, in the current experiment, the highest yield was obtained by sowing in the recommended period and applying a nitrogen dose of 200 or 150 kg ha$^{-1}$. In contrast, significantly lower yields were observed for winter wheat sown in the delayed period fertilized with 100 kg ha$^{-1}$ of nitrogen. The results were reproducible in 2021 and 2023. However, in 2022, significant differences in grain yield were observed between the recommended sowing date and the dose of 200 N kg ha$^{-1}$ compared to the delayed sowing date and the dose of 100 N kg ha$^{-1}$. Grain yields of wheat varied between the years of the study, ranging from 6.71 t ha$^{-1}$ in 2021 to 7.45 t ha$^{-1}$ in 2023 (Table 3).

**Table 3.** Grain yield in the years of research (t ha$^{-1}$).

| Sowing Date | Dose of Nitrogen (kg ha$^{-1}$) | 2021 | 2022 | 2023 | Mean over the Years |
|---|---|---|---|---|---|
| Recommended | 100 | 6.48 ± 0.21 ab | 6.59 ± 0.18 ab | 7.18 ± 0.26 ab | 6.75 ± 0.24 ab |
| | 150 | 7.34 ± 0.33 a | 7.61 ± 0.28 ab | 8.12 ± 0.37 a | 7.69 ± 0.34 a |
| | 200 | 7.62 ± 0.36 a | 7.94 ± 0.31 a | 8.41 ± 0.41 a | 7.99 ± 0.35 a |
| Mean | | 7.15 | 7.38 | 7.90 | 7.48 |
| 30-day delay | 100 | 5.52 ± 0.31 b | 5.51 ± 0.22 b | 6.12 ± 0.35 b | 5.72 ± 0.33 b |
| | 150 | 6.48 ± 0.35 ab | 6.91 ± 0.31 ab | 7.25 ± 0.28 ab | 6.88 ± 0.32 ab |
| | 200 | 6.82 ± 0.42 ab | 7.11 ± 0.35 ab | 7.61 ± 0.34 ab | 7.18 ± 0.37 ab |
| Mean | | 6.27 | 6.51 | 6.99 | 6.59 |

Results are expressed as mean value ± standard deviations. Mean values with different letters in columns are statistically different ($p < 0.05$).

Shah et al. [47] (2020) and Sobko et al. [44] demonstrated that winter wheat yield decreased by one percent with each day of delay in the sowing date, and additionally, the plants had shortened developmental stages. Fu et al. [16] reported in their study that delaying winter wheat sowing reduced yields from 6.14% to 13.72%; however, increased nitrogen fertilization alleviated this decline. The application of a higher amount of nitrogen not only increased the annual yield but also the income from cultivation despite the higher costs incurred. I showed similar results in my research.

Gheith et al. [48] demonstrated that late sowing (December) combined with low nitrogen fertilization significantly reduced the yield of the studied wheat varieties. Liang et al. [49] have concluded that excessive reduction of nitrogen fertilizers can, in the longer term, lead to a decline in soil fertility and yields. Consequently, Fixen and West [50] and Ahmed and Mahdy [51] argue that in nitrogen fertilization, it is crucial to take into account the needs of the cultivated plant, the effectiveness of nitrogen utilization, and the potential risk of negative environmental impact. It should, therefore, be stated that the results obtained in field experiments are influenced by many additional factors, including habitat conditions.

Sowing grain with a 30-day delay and applying a nitrogen dose of 100 kg ha$^{-1}$ resulted in a reduction in the protein content in the grain. A significantly higher protein content in

the grain was obtained after fertilization with a nitrogen dose of 200 kg ha$^{-1}$, regardless of the sowing date (Figure 4).

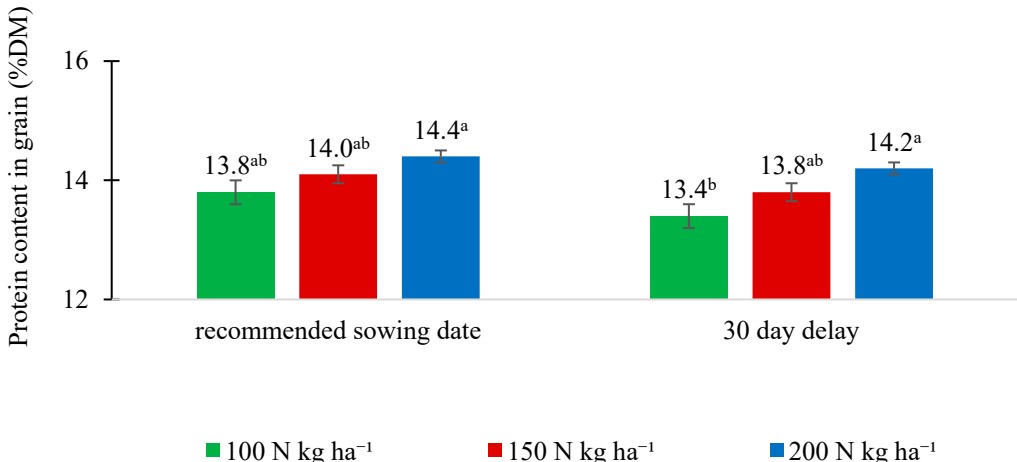

**Figure 4.** Protein content in grain in % DM—interaction between the studied factors (mean over the years). Mean values with different letters in bars are statistically different ($p < 0.05$). The standard error is marked on the bars.

Sattar et al. [52] demonstrated that delayed sowing of wheat resulted in a decreased number of grains per spike, thousand grain weight, and yield, but also in an increase in the protein content in the grain. Moreover, Shah et al. [47] and Chu et al. [9] demonstrated that delayed sowing of winter wheat increased the protein content in the grain. Mikos-Szymańska and Podolska [53] reported that variation in sowing dates had little effect on the quality of winter wheat grain, as the chemical composition of the grain was in their study primarily influenced by the genotype and harvest year. In my own research, this was confirmed, but only at the optimal sowing date. Korkhova et al. [54], on the other hand, have concluded that the quality of wheat grain is primarily influenced by weather conditions. Wojtkowiak et al. [55] have argued that the nutritional and technological value of wheat is primarily influenced by genetic traits of varieties, environmental conditions, including weather and soil, as well as agronomic practices. They demonstrated that increased nitrogen fertilization of wheat (200 kg ha$^{-1}$) had a positive impact on most evaluated quality parameters of the grain but not all. In my own research, nitrogen fertilization increases the protein content in grain, but only at a delayed sowing date.

Economic calculations showed that the application of nitrogen at 150 kg ha$^{-1}$ or 200 kg ha$^{-1}$ was beneficial but at the recommended sowing date. In each variant with a delayed sowing date, the economic effects of nitrogen fertilization were lower, especially with a dose of 100 kg ha$^{-1}$ (Table 4). But, in the case of a delayed sowing date, it is better to apply 150 kg N/ha or 200 than 100 kg N/ha.

**Table 4.** The influence of sowing date and nitrogen fertilization on economic effects.

| Sowing Date | Dose of Nitrogen (kg ha$^{-1}$) | Yield (t ha$^{-1}$) | Yield (EUR ha$^{-1}$) | Cost N (EUR ha$^{-1}$) | Economic Result |
|---|---|---|---|---|---|
| recommended | 100 | 6.75 | 1417.50 | 150.00 | 1267.5 |
| | 150 | 7.69 | 1614.90 | 225.00 | 1389.9 |
| | 200 | 7.99 | 1677.90 | 300.00 | 1377.9 |
| 30-day delay | 100 | 5.72 | 1201.20 | 150.00 | 1051.2 |
| | 150 | 6.88 | 1444.80 | 225.00 | 1219.8 |
| | 200 | 7.18 | 1507.80 | 300.00 | 1207.8 |

The grain purchase price (EUR 210/t) and nitrogen costs (EUR 1.5/kg N) were based on 2023 prices. The EUR to PLN exchange rate was 4.35.

Tabak et al. [24] have concluded that optimizing nitrogen fertilization allows high wheat yields with good grain quality, provides economic benefits, and reduces environmental risks. Keikha et al. [56] demonstrated that the application of nitrogen fertilizers at maximum levels improved the economic outcomes of wheat cultivation in both dry and humid climates compared to minimum concentrations. However, the highest nitrogen doses had a negative impact on the natural environment. Liu et al. [57] proved that nitrogen fertilization of wheat and maize had no effect on ammonium nitrogen ($NH_4$–N) content in the soil profile (except for the top 20 cm layer). However, the content of nitrate nitrogen ($NO_3$–N) underwent significant changes. The latter authors observed a strong tendency for the translocation of $NO_3$-N from the upper layer to the lower soil layers (20–100 cm) after applying doses of 240 and 360 kg N ha$^{-1}$. Velemis et al. [58] have concluded that the economic effect of nitrogen fertilization depends on various factors, including the cost of nitrogen fertilizer and the market price of grain. Therefore, economic analyses should be updated based on variable data. This was confirmed in my study.

## 4. Conclusions

Delaying the sowing dates of winter wheat (variety RGT Kilimanjaro) by 30 days compared to the recommended date in the trial area resulted in a decrease in grain yield of 0.89 t ha$^{-1}$. Weather conditions modified wheat yields in the study years, along with the effects of applied nitrogen fertilization. The difference in grain yield between 2021 and 2023 amounted to 0.74 kg ha$^{-1}$. The nitrogen dose of 200 kg ha$^{-1}$ significantly increased the Soil Plant Analysis Development (SPAD) index and Leaf Area Index (LAI) compared to the variant with delayed sowing and fertilization at a rate of 100 kg N ha$^{-1}$. The number of spikes per m$^2$ and thousand grain weight were most favorably affected by sowing grain at the recommended date and applying nitrogen at a rate of 150 or 200 kg ha$^{-1}$. The highest number of grains per spike was obtained when the grain was sown at the recommended sowing time, and the nitrogen dose was 200 kg ha$^{-1}$. Significantly lower results for the aforementioned yield components were obtained in the variant with delayed sowing and a nitrogen dose of 100 kg ha$^{-1}$. The protein content in the grain was primarily influenced by increased nitrogen fertilization, with a lesser impact on the variable sowing date. The findings of the present study allow us to conclude that postponing the sowing date of winter wheat by 30 days results in a decrease in grain yield, but this can be compensated by increased nitrogen fertilization. The most optimal economic results were obtained after applying 150 N kg ha$^{-1}$ at the recommended sowing date.

**Funding:** This research received no external funding.

**Institutional Review Board Statement:** Not applicable.

**Data Availability Statement:** The raw data supporting the conclusions of this article will be made available by the author on request.

**Acknowledgments:** I fully appreciate the editors and all anonymous reviewers for their constructive comments on this manuscript.

**Conflicts of Interest:** The author declares no conflict of interest.

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
