# Peer review of "Response of Winter Wheat to Delayed Sowing and Varied Nitrogen Fertilization"

_agriculture, doi:10.3390/agriculture14010121_

Round 1

Reviewer 1 Report

Comments and Suggestions for Authors

Dear Author,

Your paper "Response of winter wheat to delayed sowing and varied nitrogen fertilization" is interesting and might awaken the interest of readers interested in maintaining high yield and quality of grain yield in times of climate change. I found that the paper is quite well written, but also I have a few comments and suggestions that are written in the attached document. In general, the authors should improve the section with a description of the field experiment with more details (it is not clear if the experiment was carried out three years in a row on the same experimental plot or not). 

Reviewer 2 Report

Comments and Suggestions for Authors

This manuscript showed the response of the winter wheat to two sowing dates (recommended and delayed by 30 days) and three N fertilization doses, in three years field experiment. The results showed that winter wheat yield decreased at delayed sowing date and the lowest N rate as quality parameters and the economy results.

The objective is both nationally and internationally important. The abstract is informative and it covers the content of the manuscript. The discussion is relevant. Conclusions are appropriate. The author cited the relevant literature. However, factor Years is not included in a statistical analysis.

 Also, the manuscript needs a fine adjusments throughout and English reviewing. There are also writing mistakes  and use of not appropriate terms. 

  Some of the things that have to be corrected, on the way that is shown bellow are:

Line 19: ha-1

Line 21: I propose instead of »The lowest grain yield was obtained after sowing wheat at a late date and applying 100 N kg ha-1«, following: The lowest grain yield was obtained at 30-day late wheat sowing date and applying 100 N kg ha-1

Line 32: Triticum aestivum L.

Line 36: has indicated

Line 41: has reported

Line 41: significance  importance

L 134: K2O

L189: October

L189: Legend for Figure 1 is not clear. What is shown in the columns and what in the lines?

Line 191: The leaf area index (LAI) one side green area per 1 square meter of soil,

Line 222: N kg ha¹

Line 222: TGW index, showing.... (explaine what TGW is)

Line 238: Table 2. Use N instead of nitrogen

Line 285: Title of the Y osis: Protein content in grain (%DM)

Line 285: (średnia z lat) translate to English

Line 290: But, in the case of delayed sowing date, it is better to apply 150 kg N/ha or 200 than 100 kg N/ha.

Line 318: Provide better titles for the colons in the Table 8., e.g. for Yield (EUR ha-1), is it Total value? Also, what does mean Results in table 8?

Line 324: The average difference was 0.89 t ha-1???? Does not mean anything. Please provide full information.

Comments on the Quality of English Language

 The manuscript needs a fine adjusments and English reviewing. There are  a writing mistakes and use of not appropriate terms. 
